# An FPGA-Based 16-Bit Continuous-Time 1-1 MASH ΔΣ TDC Employing Multirating Technique

**Ahmad Mouri Zadeh Khaki [1], Ebrahim Farshidi [2],\*, Sawal Hamid MD Ali [3] and Masuri Othman [4]**

[1] Department of Electrical engineering, Islamic Azad University, Mahshahr Branch,
Mahshahr 6351977439, Iran; Ahmad.mouri@hotmail.com
[2] Department of Electrical engineering, Shahid Chamran University of Ahvaz, Ahvaz 6135783151, Iran
[3] Center for Integrated Systems Engineering and Advanced Technologies, Faculty of Engineering and Built
Environment, National University of Malaysia, Bangi 43600, Malaysia; Sawal@ukm.edu.my
[4] Institute of Microengineering and Nanoelectronics, National University of Malaysia, Bangi 43600, Malaysia;
Masuri@ukm.edu.my
\* Correspondence: Farshidi@scu.ac.ir; Tel.: +98-916-614-8163

**Abstract:** An all-digital voltage-controlled oscillator (VCO)-based second-order multi-stage noise-shaping (MASH) ΔΣ time-to-digital converter (TDC) is presented in this paper. The prototype of the proposed TDC was implemented on an Altera Stratix IV FPGA board. In order to improve the performance over conventional TDCs, a multirating technique is employed in this work in which higher sampling rate is used for higher stages. Experimental results show that the multirating technique had a significant influence on improving signal-to-noise ratio (SNR), from 43.09 dB without multirating to 61.02 dB with multirating technique (a gain of 17.93 dB) by quadrupling the sampling rate of the second stage. As the proposed design works in the time-domain and does not consist of any loop and calibration block, no time-to-voltage conversion is needed which results in low complexity and power consumption. A built-in oscillator and phase-locked loops (PLLs) of the FPGA board are utilized to generate sampling clocks at different frequencies. Therefore, no external clock needs to be applied to the proposed TDC. Two cases with different sampling rates were examined by the proposed design to demonstrate the capability of the technique. It can be implied that, by employing multirating technique and increasing sampling frequency, higher SNR can be achieved.

**Keywords:** delta-sigma modulation; multirating technique; multi-stage noise-shaping (MASH); time-to-digital converter (TDC); voltage-controlled oscillator (VCO)

---

## 1. Introduction

In many applications such as all-digital phase-locked loops (ADPLLs) [1], chemical sensors readout [2], frequency synthesizers [3–6], and time-of-flight (ToF) systems [7], time-to-digital converters (TDCs) play an important role by measuring a time interval. Thus far, many TDCs have been presented which have been trying to show high signal-to-noise ratio (SNR), resolution, bandwidth, and linearity. In this way, various TDC architectures such as time-interleaved, pipelined, flash, Successive approximation register (SAR) and cyclic architectures have been introduced [8–14]. Since these architectures operate in Nyquist rate, they are ineligible to achieve important parameters of performance such as dynamic range and resolution higher than that of their oversampling counterparts.

Recently, researchers have introduced ΔΣ TDCs benefiting from an inherent noise-shaping property. Voltage-domain ΔΣ TDCs which are implemented mainly as analog utilize a time-to-voltage converter and a conventional ΔΣ modulator [15,16]. To take advantage of the scaling of the CMOS process, time-domain ΔΣ TDCs utilizing digital circuits such as multi-bit counter, gated-ring oscillator

(GRO) [17] and switched-ring oscillator (SRO) [18] have been proposed. However, the noticeable problem of GRO-TDC is the oversampling ratio (OSR) limitation by the rate of input pulse (*fc*). Also, a SRO-TDC suffers from noise-shaping limitation to the first order. Consequently, achieving fine time-resolution is not affordable in these architectures. However, to achieve finer time-resolution a 1-1 multi-stage noise-shaping (MASH) TDC can be formed by cascading two SRO-TDCs using two identical SROs, and the difference in the SROs operating frequencies results in systematic error which requires calibration unit to compensate the error. Therefore, this takes a long settling time and also suffers from additional power consumption and chip area [19]. A second-order MASH $\Delta\Sigma$ TDC has been presented in [20] which solves the aforementioned problem in a 1-1 MASH SRO-TDC by exploiting gated switched-ring oscillators (GSROs) that lead to a fine time-resolution without calibration. However, the main drawback of that work is OSR limitation due to GSROs' operating frequencies. As will be discussed later, in GSRO-TDC a counter counts the number of rising edges of GSRO output which is proportional to the time interval to be measured. Thus, sampling clock frequency (*fs*) must be less than maximum frequency of the GSRO so that at least one rising edge occurs in each sampling clock. Therefore, as operating frequencies of the GSRO increase, higher *fs* can be applied to the TDC which results in higher time-resolution. To enhance SNR, a multirated 1-1 MASH $\Delta\Sigma$ TDC has been proposed in [21]. Although it exploits a digital ring oscillator in the second stage, a switched-capacitor VCO is used in its first stage that not only suffers from non-linearity and low operating frequency but also because of analog implementation occupies high chip area and consumes excessive power. Moreover, the second-stage ring oscillator produces a thermometric code corresponding to its input. Thus, an extra unit is needed to decode the thermometric code to a binary counterpart resulting in more chip area and power consumption. Also, the aforementioned decoding takes three clock cycles to be completed which degrades the speed of TDC. In the meantime, advancing of CMOS technology on the one hand, and introducing high-performance FPGA chips on the other, have had an impressive impact on presenting fast, accurate and low power application specific integrated circuit (ASIC) and FPGA-based TDCs [22–29]. However, all-digital designing encourages some designers to implement their works on FPGA. Albeit optimizing the design in ASIC implementation, the benefit of FPGA implementation is that the design can be programmable to suit different applications. So, it can be tailored for wide variety of applications compared with ASIC which is not flexible. Although the FPGA boards are too big, the board is only used during the development phase. Once the design is confirmed, only the FPGA chip will be embedded in the system rather than the whole development board. Hence, we can say that the FPGA chip size is almost similar to ASIC IC while it can perform multiple operations and flexibility.

This paper presents an FPGA-based 1-1 MASH GSRO-based $\Delta\Sigma$ TDC which employs multirating technique in a $\Delta\Sigma$ GSRO-TDC for the first time. We propose a 16-bit continuous-time TDC that takes advantage of employing GSRO quantizer to suppress quantization error leakage and at the same time benefits from a multirating technique to improve SNR further over conventional TDCs. The operation principle and FPGA implementation are described in detail. The proposed FPGA-based TDC achieves a high performance in terms of dynamic range, time-resolution and figure-of-merit (FoM) while providing acceptable SNR and bandwidth.

This paper is organized as follows. In Section 2, a background of GSRO-based MASH $\Delta\Sigma$ TDCs and multirating technique is presented. The proposed multirated 1-1 MASH $\Delta\Sigma$ TDC is introduced in Section 3. Section 4 describes the implementation details of the proposed TDC. In Section 5, the measured results of the FPGA-based prototype TDC are discussed. Finally, Section 6 concludes this paper.

## 2. Background

A remarkable reason for popularity of $\Delta\Sigma$ TDCs is their inherent property of noise-shaping. Moreover, as the order of noise-shaping increases the noise power in the band of interest decreases [24]. Therefore, a higher order of noise-shaping is of interest. Increasing the order of noise-shaping is

achievable either by increasing loop order or cascading a higher number of stages (multi-stage noise-shaping (MASH)) which both result in more complexity and subsequently more power consumption and chip area. Hence, there is a trade-off in designing $\Delta\Sigma$ TDCs in terms of performance and complexity. Thus, by choosing a proper order of noise-shaping and maximizing the improvement of critical specifications of TDC, acceptable performance can be gained. This section describes the theoretical background of $\Delta\Sigma$ TDCs using a GSRO and the multirating technique to give an overview of the proposed architecture. This theoretical background will be based on [17] as our benchmark.

### 2.1. Gated Switched-Ring Oscillator-Time-to-Digital Converter (GSRO-TDC)

As mentioned previously, due to frequency differences between SROs in a 1-1 MASH structure, a SRO-SRO TDC suffers from phase-domain quantization error leakage and cannot achieve second-order noise shaping properly. In order to alleviate this problem, GSRO can be employed. It should be noted that GSRO is obtained by adding phase-holding gates to the supply and ground of an SRO. Figure 1 illustrates the operation principle of a GSRO. As shown, when the gates are closed GSRO acts the same as a SRO and when the gates are open it holds its phase. Basically, GSRO provides three operating frequencies: $f_{max}, f_{min}$ and 0. By keeping phase properly in GSRO, leakage can be avoided effectively.

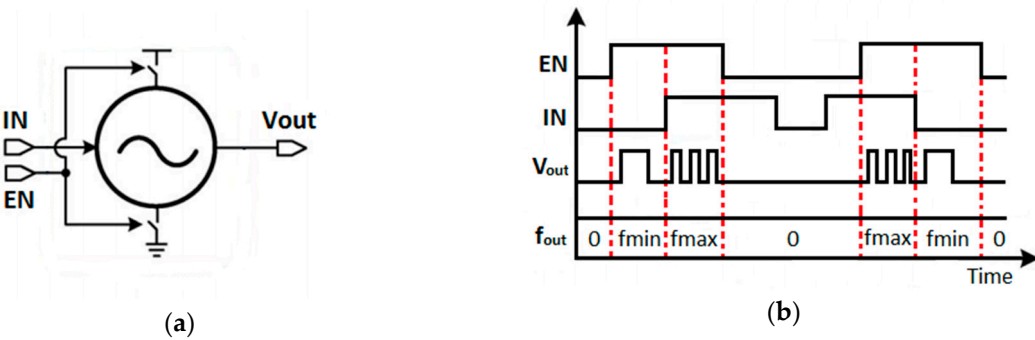

**Figure 1.** Gated switched-ring oscillator (GSRO): (**a**) block diagram; (**b**) waveforms [20].

To clarify superiority of GSRO-TDC over SRO-TDC in suppressing quantization error leakage, we compare their output relationships. While the output of SRO-TDC in z-domain is given by:

$$2\pi D_{OUT} = z^{-1}D_1 - (1-z^{-1})D_2 \tag{1}$$

$$\begin{aligned}= \; & z^{-1}\phi_{SRO1} - z(1-z^{-1})^2\phi_{Q2} \\ & +(1-z^{-1})\left(1-\frac{\phi_{SRO2}}{\phi_{Q1}}\right)\phi_{Q1}\end{aligned} \tag{2}$$

where the phase-domain quantization error of the first stage and the phase change of the second stage are described by:

$$\phi_{Q1} = 2\pi \int_{T_{Q1}} f_{SRO1}\, dt \tag{3}$$

$$\phi_{SRO2} = 2\pi \int_{T_S} f_{SRO2}\, dt. \tag{4}$$

Yet, when second stage of TDC exploits a GSRO, the phase change of the second stage GSRO ($\phi_{GSRO2}$) can be estimated by:

$$\begin{aligned}\phi_{GSRO2}[n] &= 2\pi \int_{T_S} f_{GSRO2}\, dt \\ &= 2\pi \int_{T_{Q1[n]}} f_{GSRO2}\, dt = \phi_{Q1}[n].\end{aligned} \tag{5}$$

As evidenced by Equation (5), by exploiting a GSRO in the second stage, because of the equality of Equations (3) and (4) leakage can be suppressed perfectly. Therefore, by contrast with the SRO-SRO MASH, a GSRO-TDC does not need calibration and can achieve second-order noise shaping efficiently [20]. A GSRO-based 1-1 MASH TDC has been introduced in [20] the block diagram of which is shown in Figure 2. The input pulse is produced from Start and Stop signals by PulseGen. By closing the Enable (EN) gates of the GSRO1, it is configured as an SRO. Thus, the first stage operates as a conventional SRO-TDC. The input pulse, sampling clock, and output of the GSRO1 ($Y_1$) are fed to QEGen that produces a quantization error pulse of the first stage ($Q_1$) and a frequency sync pulse ($Q_{IN}$) in every cycle which control the inputs of the second stage. Since the gates of the GSRO2 are controlled by $Q_1$ and the frequency of that is controlled by $Q_{IN}$, oscillation frequencies of the GSRO1 and GSRO2 are equal during a sampling period as shown in Figure 3.

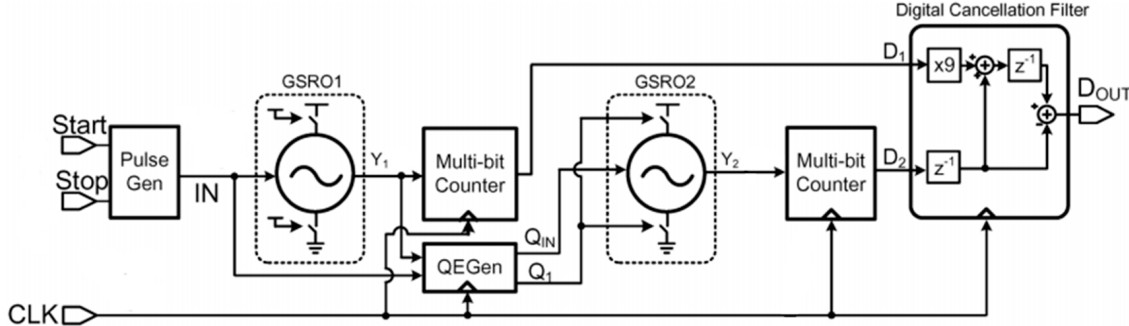

**Figure 2.** 1-1 Multi-stage noise-shaping (MASH) $\Delta\Sigma$ time-to-digital converter (TDC) architecture using GSRO [20].

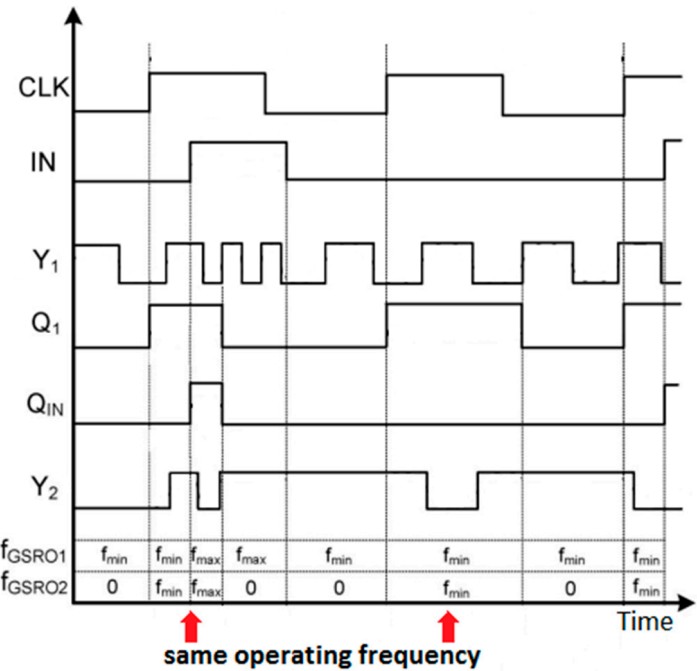

**Figure 3.** The timing diagram of the second-order GSRO-TDC [20].

It is worth mentioning that since in order to accomplish second-order noise-shaping a residue pulse must be generated in each cycle, the frequency of the GSRO is designed to be higher than $f_S$ so that there exists at least one rising edge during a sampling period. Therefore, the relationship between the GSRO frequency, $f_S$ and $f_C$ can be expressed as: $f_{max} > f_{min} > f_S \geq f_C$ [20]. Thus, it can be said that in GSRO-TDC OSR is limited by the frequency of GSRO. So, we set $f_{max}$ and $f_{min}$ of GSROs in this work at

4 GHz and 2 GHz, respectively, so that they are 1.6 and 2 times their counterpart in [20]. Such high GSRO frequency makes applying $f_S$ up to 2 GHz eligible in this design which results in higher OSRs and hence time-resolution below 0.5 ps.

### 2.2. Multirating Technique in $\Delta\Sigma$ Multi-Stage Noise-Shaping (MASH) Converters

Despite the fact that increasing OSR results in enhancement of SNR [30], increasing sampling frequency imposes more power consumption on the circuit. As will be described, the quantization noise error of the first stage is removed by a digital cancelation filter (DCF) and has a negligible influence on the overall performance of the TDC. Thus, we apply a sampling frequency the same as conventional $\Delta\Sigma$ TDCs to this stage to save power budget while we increase sampling frequency of higher stages to improve SNR. Adjusting sampling clocks of various stages at different frequencies independently in $\Delta\Sigma$ converters is called the multirating technique. Employing this technique helps the designer to improve performance of converter by expanding the design space. However, the main aim of employing this technique is SNR enhancement in $\Delta\Sigma$ converters via increasing OSR while trying to prevent consuming impressive power. Figure 4 depicts employing the multirating technique in a 1-1 MASH GSRO-TDC. As can be seen, the first stage operates at a frequency ($f_{S1}$) and second stage operates at higher speed ($f_{S2} = m \times f_{S1}$). As a matter of fact, employing multirating technique is more attractive in $\Delta\Sigma$ MASH converters rather than single-loop converters, because in the MASH architecture each stage operates independently and thus no interstage feedback is required while the latter require complex feedback, multi-bit digital-to-analog converter (DAC) and anti-aliasing digital filter. Also, as op-amp based and analog implemented converters suffer from gain/bandwidth trade-off and limited speed, performance improvement by employing this technique is limited in such converters [21]. Thus, an all-digital multirate $\Delta\Sigma$ MASH TDC that can achieve high performance is of interest. To reveal maximum performance enhancement by exploiting the multirating technique at high clock frequencies, this paper presents an FPGA-based 1-1 MASH $\Delta\Sigma$ TDC employing the multirating technique in a GSRO-TDC for the first time.

A multirate MASH TDC principally operates the same way as conventional single-rate MASH structure. At the front end, the output of the individual stages is combined by the DCF so that only the last stage quantization error remains. The noise transfer function (NTF) whose order is the sum of the order of the stages shapes the overall quantization noise error. The overall NTF of a general N-stage multirated MASH structure ($\text{NTF}_{\text{MR}}$) is expressed as:

$$\text{NTF}_{\text{MR}} = \prod_{1}^{\text{N}} \left(1 - z^{-\frac{OSR_1}{OSR_i}}\right)^{n_i} \tag{6}$$

where $n_i$ is the loop filter order and $OSR_i$ is over-sampling-ratio (OSR) of the i[th] stage. The NTF of conventional single-rate MASH structure is obtained by:

$$\text{NTF}_{\text{SR}} = \prod_{1}^{\text{N}} \left(1 - z^{-1}\right)^{n_i} \tag{7}$$

Comparing Equation (6) with Equation (7) reveals that the SQNR improvement in the multirated case compared to single-rate structure is given by:

$$\text{SQNR}_{\text{MR-SR}} = \sum_{i=1}^{i=N-1} \left(2n_i 10\log_{10}\left(\frac{OSR_i}{OSR_1}\right)\right) + (2n_N + 1)10\log_{10}\left(\frac{OSR_N}{OSR_1}\right) \tag{8}$$

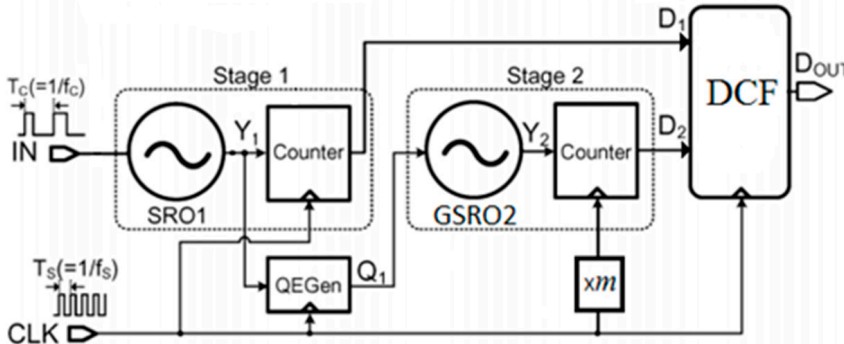

**Figure 4.** Block diagram of the 1-1 MASH multirate structure.

## 3. Proposed FPGA-Based 1-1 MASH $\Delta\Sigma$ TDC

Considering the advantages of the GSRO-TDC and multirating technique in the previous section, a 16-bit FPGA-based 1-1 MASH $\Delta\Sigma$ TDC using identical GSROs in either the first or second stages improved by the multirating technique is introduced in this work to enhance performance further rather than conventional GSRO-TDCs. The block diagram of the proposed TDC is shown in Figure 5. Comparing this figure to Figure 2, it can be realized that this work utilizes the same structure as the TDC introduced in [20] but the sampling frequency of the second stage ($f_{S2}$) is multiplied by the multirating ratio ($m$) in the proposed TDC. In addition, as $f_{S2}$ is $m$ times higher than the first stage clock frequency ($f_{S1}$), output of the first stage ($Y_1$) should be up-sampled by $m$ in the DCF. Like conventional $\Delta\Sigma$ converters, the first- and second-stage outputs ($Y_1$ and $Y_2$) should be filtered by the signal transfer function (STF) of the second stage ($z^{-1}$) and NTF of the first stage ($1 - z^{-1}$), respectively. Finally, the overall output of the proposed TDC ($D_{OUT}$) is obtained by subtraction of $Y_2$ from $Y_1$ which can be expressed as:

$$
\begin{aligned}
D_{OUT} &= z^{-1}Y_1 - (1 - z^{-1})Y_2 \\
&= \tfrac{1}{2\pi}\left(\phi_{Q1}(z) - z(1 - z^{-1})^2 \phi_{GSRO2}(z)\right)
\end{aligned}
\tag{9}
$$

From Equation (9), it is obvious that second-order noise-shaping can be achieved by the proposed $\Delta\Sigma$ TDC.

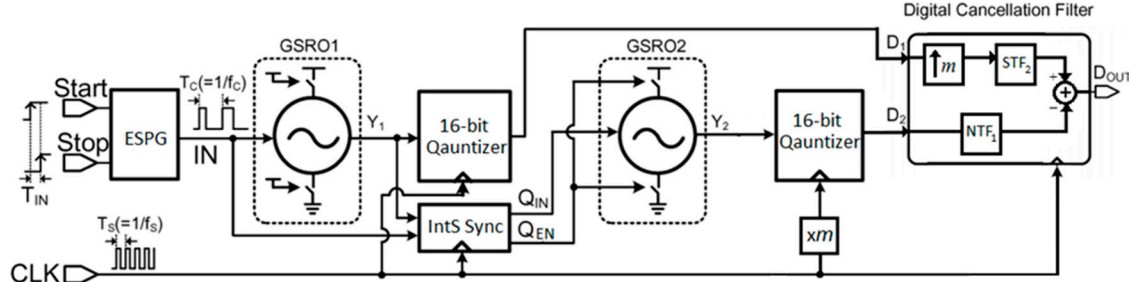

**Figure 5.** Block diagram of the proposed 1-1 MASH multirate TDC.

To illustrate the operation of the proposed TDC, the timing diagram of that for 1 ns and 4 ns input time interval at 100 MHz input pulse, 200 MHz and 800 MHz $f_{S1}$ and $f_{S2}$ are shown in Figures 6a and 6b, respectively. As shown, a quantization error pulse of the first stage ($Q_{EN}$) is generated by an interstage synchronizer (IntS Sync) every cycle to enable GSRO2 and ($Q_{IN}$) to synchronize the GSROs operating frequency. Hence, as can be seen, this approach suppresses any frequency difference between GSROs perfectly which removes phase-domain quantization error leakage that allows the proposed design to achieve second-order noise-shaping as expected. Moreover, it can be deduced that the overall output number related to input time interval ($D_{OUT}$) is the average of numbers counted in each sampling clock ($CNT_{OUT}$).

To demonstrate the operation of the proposed TDC at higher frequencies, 200 MHz input pulse, 400 MHz and 1.6 GHz sampling clocks were applied to the first and second stages, respectively, the timing diagram of which is shown in Figure 6c,d for 1 ns and 2 ns input interval. Figure 6 implies that the proposed design can operate properly as a $\Delta\Sigma$ TDC and can be employed in high-speed applications requiring this unit such as ADPLLs and frequency synthesizers.

As mentioned previously, by using the GSRO in both first and second stages, phase-domain quantization error leakage can be avoided effectively which results in higher SNR. Moreover, benefiting from a high-speed Altera Stratix IV FPGA board, $f_{min}$ and $f_{max}$ of the GSROs are set at higher frequencies than state-of-the-art VCO-based $\Delta\Sigma$ TDCs that allow OSRs greater than that feasible in previous works which result in finer time-resolution. Yet, by exploiting multirating technique, a notable improvement in performance of the proposed TDC rather than previous works using same architecture is attainable. Due to the speed and power limitation in analog circuitries used in the first stage of 1-1 MASH TDC in [21], it cannot benefit from this technique at high frequencies to enhance performance further. Hence, fine time-resolution below 1 ps is not achievable in that work. In addition, while in other works $f_{S1}$ and $f_{S2}$ are applied to the TDC using external sources, in the proposed design built-in PLLs of the Altera Stratix IV FPGA board provide $f_{S1}$ and $f_{S2}$ and, accordingly, no external sources are needed.

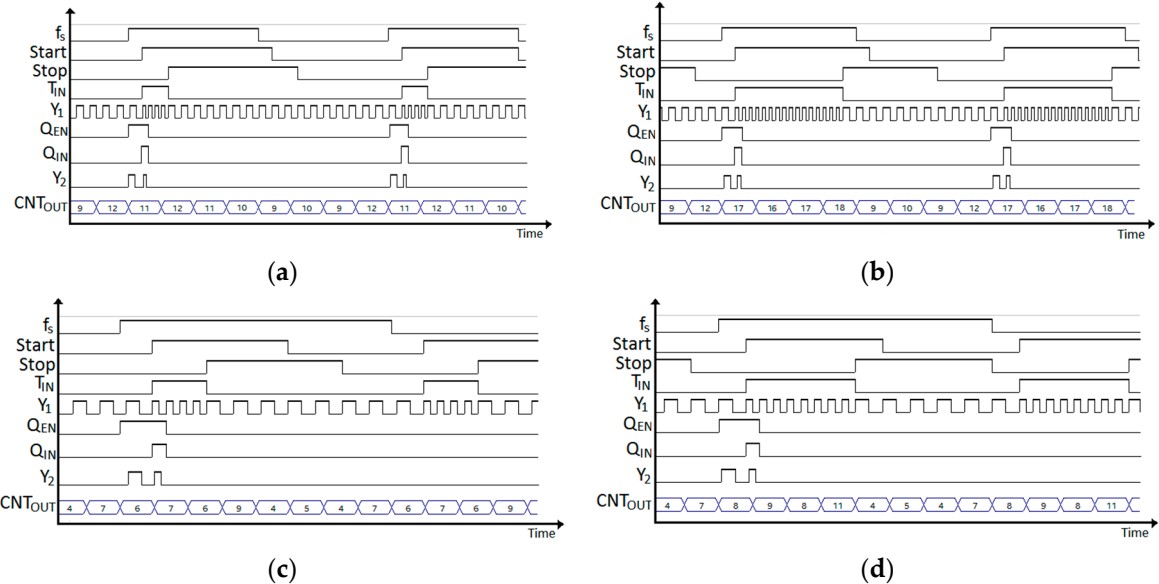

**Figure 6.** Timing diagram of the proposed 1-1 MASH multirate TDC: (**a**) $f_{S1}$ = 200 MHz and $f_{S2}$ = 800 MHz with $T_{IN}$ = 1 ns; (**b**) $f_{S1}$ = 200 MHz and $f_{S2}$ = 800 MHz with $T_{IN}$ = 4 ns; (**c**) $f_{S1}$ = 400 MHz and $f_{S2}$ = 1600 MHz with $T_{IN}$ = 1 ns; (**d**) $f_{S1}$ = 400 MHz and $f_{S2}$ = 1600 MHz with $T_{IN}$ = 2 ns.

## 4. Implementation Details

### 4.1. GSRO and Sampling Clocks

A considerable advantage of the proposed FPGA-based TDC is employing built-in oscillator and PLLs of the Altera Stratix IV FPGA board for providing different clocks required for high-performance TDC operation. For this purpose, a built-in 100 MHz crystal oscillator and PLLs are utilized from which different frequencies for sampling clocks ($f_{S1}$ and $f_{S2}$) and GSROs operation frequencies ($f_{min}$ and $f_{max}$) are extracted. Figure 7a illustrates the conceptual clock generating different frequencies in the proposed design. As can be seen, in order to obtain $f_{S1}$, $f_{S2}$, $f_{min}$ and $f_{max}$, 100 MHz input clock is multiplied by 2, 8, 20 and 40, respectively. Furthermore, as shown in Figure 7b, a control unit with 3-state output emulates GSRO. When both $Q_{EN}$ and $Q_{IN}$ are low, the unit output is 0. If $Q_{EN}$ is high but $Q_{IN}$ is low $f_{min}$ goes to output. When both $Q_{EN}$ and $Q_{IN}$ are high output is $f_{max}$.

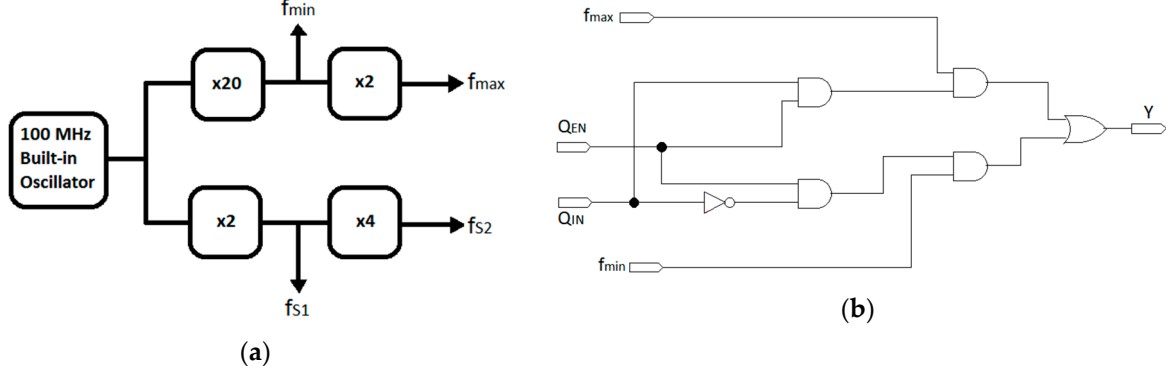

**Figure 7.** (**a**) Conceptual clock generating unit in the proposed TDC; (**b**) schematic of GSRO operating frequency control unit.

Again, it is worth saying that in the proposed FPGA-based TDC no external sources required to perform sampling clocks which eliminates the need of a discrete clock generator resulting in easier measurement, because only input interval pulses should be applied to the proposed TDC. Also, taking advantage of high-speed Altera Stratix IV FPGA board implemented in 40 nm technology, 2 GHz and 4 GHz clocks can be provided easily to be used as $f_{min}$ and $f_{max}$ which result in better time-resolution compared to previous GSRO-TDCs with lower operating frequencies for GSRO, but high switching delay makes it difficult to obtain such high frequencies by using analog circuitries and conventional ring oscillators.

*4.2. Interstage Synchronizer (IntS Sync)*

In each cycle, IntS Sync produces $Q_{EN}$ to enable GSRO2 and $Q_{IN}$ to synchronize operating frequency of two GSROs. Schematics of this unit and timing diagram of that are shown in Figures 8a and 8b, respectively. An edge-sensitive pulse generator (ESPG) of that schematic which is shown in Figure 8c produces $Q_{EN}$ with the width of interval between rising edges of CLK and $Y_1$. Then $Q_{EN}$ which is proportional to the quantization error of the first stage is fed to GSRO2. This pulse can be very narrow and can even be ignored if rising time of the ESPG is larger than the quantization error pulse width. Thus, owing to limited GSRO gates switching time, a narrow pulse causes a deadzone problem and degrades the operation of the proposed TDC. Nevertheless, by adding a flip-flop (DFF2) a static offset of $2\pi$ is added to $Q_{EN}$ to avoid this problem. The static offset is removed in the DCF and the overall performance of the proposed TDC is not altered [20].

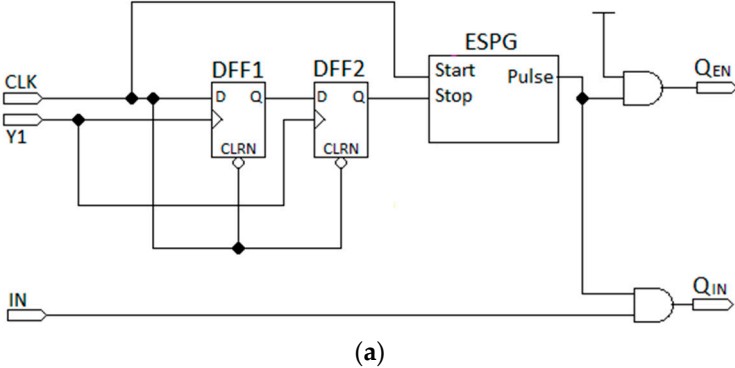

(**a**)

**Figure 8.** *Cont.*

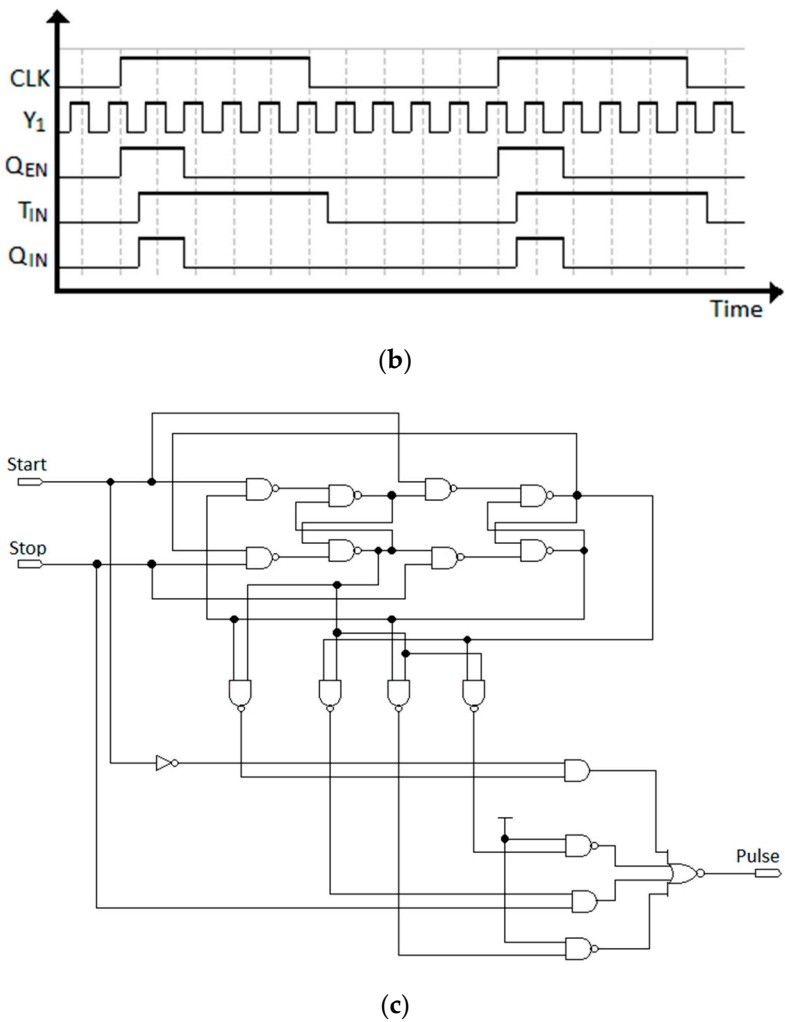

(**b**)

(**c**)

**Figure 8.** Interstage synchronizer (IntS Sync): (**a**) block diagram; (**b**) timing diagram; (**c**) schematic of the edge-sensitive pulse generator (ESPG).

*4.3. The 16-Bit Quantizer*

As described in Section 2, designer must set $f_S$ lower than $f_{min}$ to guarantee occurring a residue pulse each cycle. As $f_{min} > f_S$, a number of rising edges may appear in a sampling period and, thus, a multi-bit quantizer is needed. In the proposed FPGA-based TDC, built-in 16-bit counters count rising edges of the first and second stages outputs to quantize $Y_1$ and $Y_2$ in each sampling clock. Using such number of bits for quantization is a notable distinction point of the proposed TDC compared to previous works results in a better time-resolution. It is noteworthy that such an improvement is an advantage of implementing the proposed all-digital TDC on a high-performance FPGA board. A possible error that may occur during 16-bit quantizer operation is coinciding the transition of counter output with a sampling clock rising edge [31]. To avoid this large error, a delayed clock generator (DLCKgen) shown in Figure 9a is employed so that sampling clock occurs only after settling counter output ($CNT_{out}$). As shown in Figure 9b, such an error is reduced effectively by adding a DLCKgen [20].

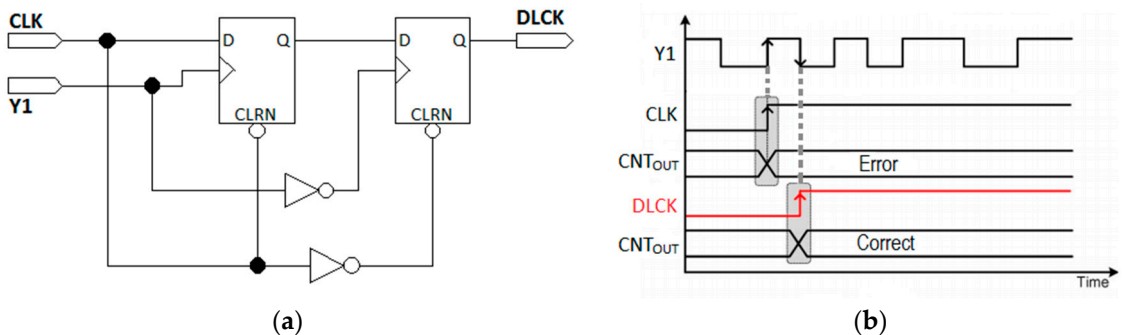

**Figure 9.** Delaying rising edges of Y2 to avoid error: (**a**) schematic of delayed pulse generator; (**b**) timing diagram.

### 4.4. Digital Cancellation Filter (DCF)

In order to achieve second-order noise-shaping, digital output of two stages are combined in the DCF. The proposed TDC utilizes an on-chip DCF to remove quantization noise error of the first stage. As shown in Figure 5, the first-stage digital output is up-sampled by multirating ratio 4 and then filtered by the STF of the second stage ($z^{-1}$) to produce *D1*. Also, digital output of the second stage is filtered by the NTF of the first stage ($1 - z^{-1}$) to generate *D2*. Finally, the overall TDC digital output ($D_{OUT}$) is obtained by subtraction of *D2* from *D1*.

Taking advantage of all-digital designing, DCF of the proposed TDC is implemented on-chip using built-in provided logic gates and arithmetic units of the Altera Stratix IV FPGA board, while it is implemented off-chip and the STF of the second stage needs to be tuned manually to achieve maximum SNDR in [21].

## 5. Measured Results

The proposed design was implemented on an Altera Stratix IV FPGA board and general purpose input/output (GPIO) port of that is used to apply input time interval ($T_{IN}$) and obtain digital output ($D_{OUT}$). It should be mentioned that the I/O standard of GPIO ports of the Altera Stratix IV FPGA board is 3 V. Thus, in order to obtain output spectrum, 3 V 10 MHz input pulses are applied to the proposed TDC using a function generator (Siglent SDG 1050) and 100-k samples are captured using a mixed-domain oscilloscope (Tektronix MDO 4104) that uses a Hann window. Then, post-processing of captured data was done using MATLAB. The measurement setup for obtaining the output spectrum of the proposed TDC is illustrated in Figure 10.

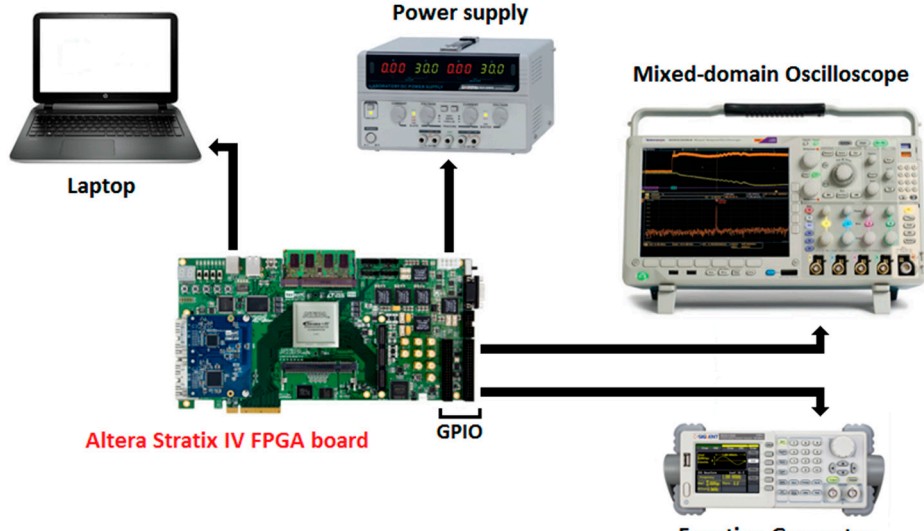

**Figure 10.** The measurement setup for obtaining the output spectrum of the proposed 1-1 MASH TDC.

First, the proposed TDC was examined in single-rate case to verify the ability of achieving second-order noise-shaping. Figure 11a shows the measured output spectrum of the proposed TDC in a single-rate case at 100 MHz $f_C$ and 200 MHz $f_{S1}$ and $f_{S2}$. As shown, this work can achieve appropriate second-order noise-shaping, and the measured SNR within 9.5 MHz bandwidth is 43.09 dB in this case. To demonstrate the improvement resulting from the multirating technique, a sampling clock of the second stage was quadrupled and the output spectrum of the proposed TDC at 100 MHz $f_C$, 200 MHz $f_{S1}$ and 800 MHz $f_{S2}$ is shown in Figure 11b. As surmised, benefiting from multirating technique the measured SNR within 9.6 MHz bandwidth was 56.8 dB which showed 13.71 dB enhancement compared to single-rate case. According to Equation (8), by increasing the OSR, more enhancement in SNR is achievable. Therefore, another measurement was induced to the proposed TDC with higher $f_{S1}$ and $f_{S2}$. In this case, 400 MHz $f_{S1}$ and 1.6 GHz $f_{S2}$ were applied to the first and second stages, respectively.

By applying 100 MHz $f_C$ to the proposed TDC, the output spectrum was measured and the result of which was shown in Figure 11c. Interestingly, yielding 4.22 dB enhancement rather the previous case, the measured SNR within 9.6 MHz bandwidth was 61.02 dB which translates to 9.8 effective number of bits (ENOB) and 0.27 ps time-resolution. The core power consumption of the Altera Stratix IV FPGA board is 6.23 mW at 100 MHz $f_C$, 200 MHz $f_{S1}$ and 800 MHz $f_{S2}$. By increasing the sampling frequency (i.e., 400 MHz $f_{S1}$ and 1.6 GHz $f_{S2}$), the power consumption increases up to 7.84 mW while it not only results in better time-resolution and SNR, but also increases the figure-of-merit (FoM) from 176 dB to 177 dB.

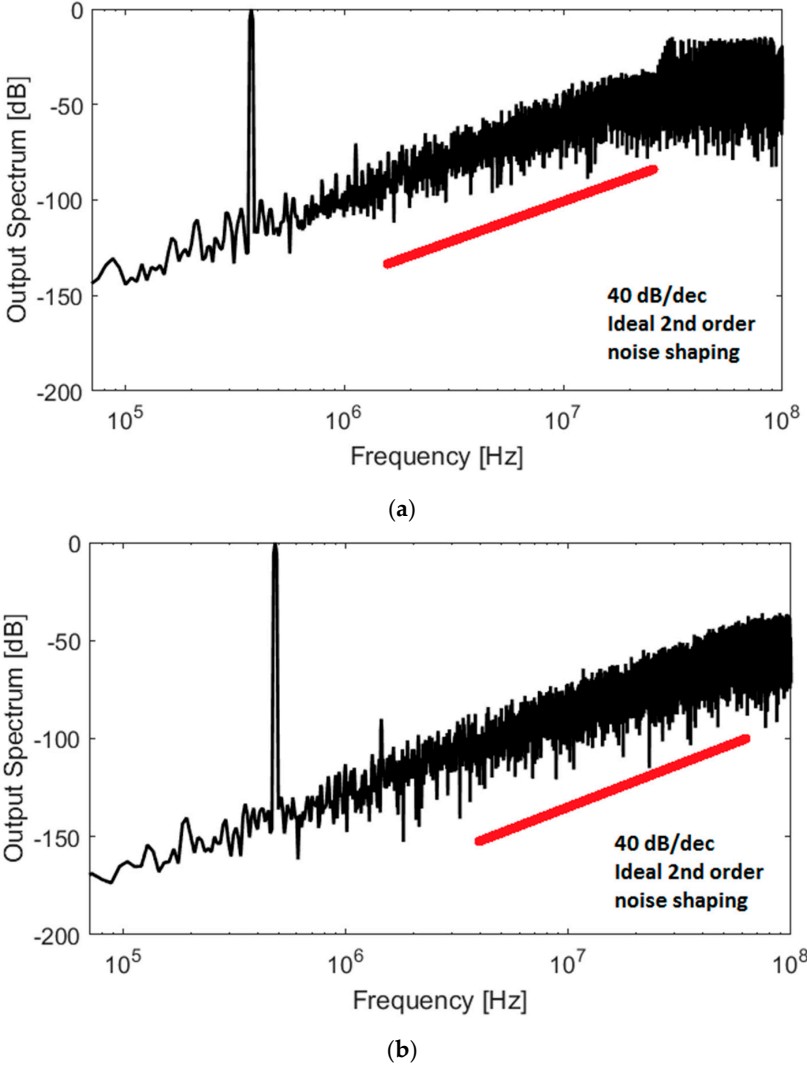

(a)

(b)

**Figure 11.** *Cont.*

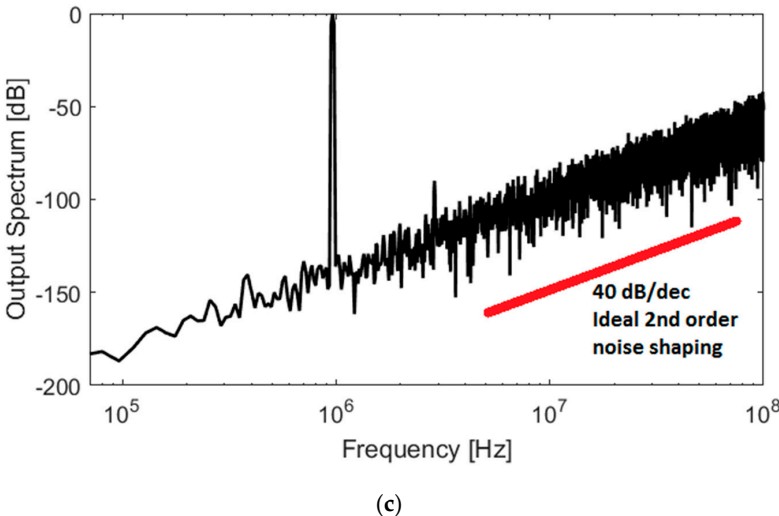

(**c**)

**Figure 11.** Measured output spectrum of the proposed 1-1 MASH TDC. (**a**) $f_{S1}$ = 200 MHz and $f_{S2}$ = 200 MHz; (**b**) $f_{S1}$ = 200 MHz and $f_{S2}$ = 800 MHz; (**c**) $f_{S1}$ = 400 MHz and $f_{S2}$ = 1600 MHz.

Increasing $f_C$ leads to higher SNR for the same OSR [20]. So, we examined this effect by the prototype of the proposed TDC once at 200 MHz $f_{S1}$ and 800 MHz $f_{S2}$, and another time at 400 MHz $f_{S1}$ and 1.6 GHz $f_{S2}$. Figure 12 depicts the measured SNR versus $f_C$ while OSR remains unchanged. According to Figure 12, it can be deduced that the SNR is improved by increasing $f_C$ for both cases.

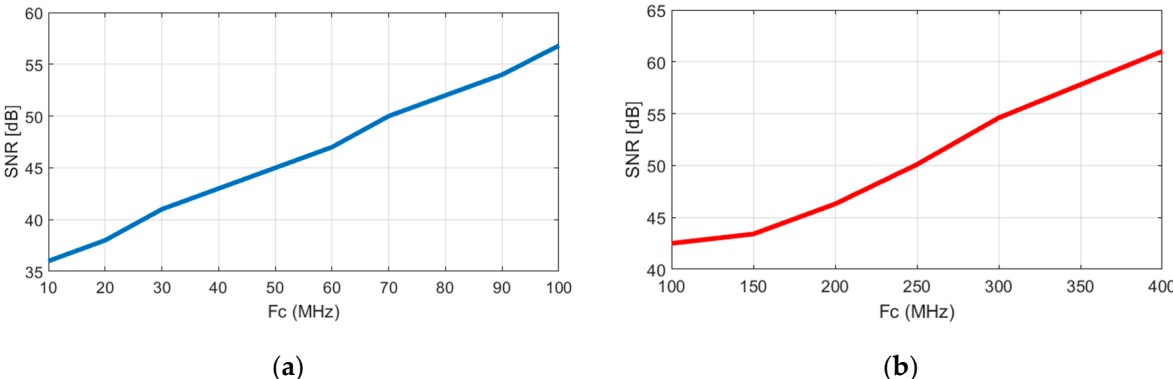

(**a**)                                                           (**b**)

**Figure 12.** Signal-to-noise ratio (SNR) versus $f_C$: (**a**) $f_{S1}$ = 200 MHz and $f_{S2}$ = 800 MHz; (**b**) $f_{S1}$ = 400 MHz and $f_{S2}$ = 1.6 GHz.

Table 1 summarizes the performance of the proposed TDC and compares it with the state-of-the-art $\Delta\Sigma$ TDCs. Owing to all-digital designing, employing the multirating technique and implementation on a high-speed FPGA board, the proposed TDC exhibited superior time-resolution, dynamic range and FoM than in previous works. Moreover, Table 1 reveals that this work represents an acceptable SNR compared with TDCs in [23–25] while they utilize higher order of noise-shaping, and higher SNR over the same order TDC in [22]. Therefore, we can surmise that higher SNR can be attained by increasing noise-shaping order via incorporating more stages to the proposed structure.

**Table 1.** Performance summary and comparison with other state-of-the art $\Delta\sum$ TDCs.

| | [25] | [24] | [23] | [22] | [20] | This work | |
|---|---|---|---|---|---|---|---|
| Process (nm) | 40 | 40 | 40 | 65 | 65 | 40-FPGA | |
| Shaping order | 4 | 3 | 3 | 2 | 2 | 2 | |
| $f_{BW}$ (MHz) | 50.3 | 156 | 50.5 | 2.5 | 4 | 9.6 | |
| $T_{range}$ (ns) | N/A | N/A | N/A | 4.5 | 4 | 4.5 | |
| $f_S$ (MHz) | 1000 | 5000 | 3000 | 205 | 400 | 200 - 800 | 400 - 1600 |
| DR (dB) | 76.8 | 70 | 68.2 | 52.6 | 79.6 | 84.2 | 86.2 |
| $T_{int,rms}$ ($f_{s,rms}$) [1] | N/A | N/A | N/A | 3752 | 148 | 98.2 | 78 |
| SNR (dB) | 75.8 | 66.6 | 68 | 56 | N/A | 56.8 | 61.02 |
| Resolution (ps) [2] | N/A | N/A | N/A | 13 | 0.51 | 0.34 | 0.27 |
| Power (mW) | 43 | 233 | 19 | 0.63 | 6.72 | 6.23 [3] | 7.84 [3] |
| FoM (dB) [4] | 167.5 | 158.3 | 162.4 | 148 | 167 | 176 | 177 |

[1] Estimated integrated noise ($\sqrt{Resolution^2/12}$). [2] Estimated resolution ($\sqrt{T_{int,rms}^2.12}$). [3] FPGA core power consumption.
[4] FoM = DR + 10 $\log_{10}$ (Bandwidth/Power) [dB], where $DR = 20 \log_{10} (T_{range,rms}/T_{int,rms})$.

## 6. Conclusions

In this paper, a novel 16-bit second-order $\Delta\sum$ TDC has been presented. The proposed structure exploits GSRO quantizers in either first or second stages which quantize input pulses in the time domain using MASH architecture. The proposed design employs a multirating technique for the first time in a GSRO-TDC to improve performance over state-of-the-art TDCs. The prototype of the proposed TDC is implemented on an Altera Stratix VI FPGA board and achieves 61.02 dB SNR within 9.6 MHz bandwidth with 400 MHz and 1.6 GHz sampling frequencies in the first and second stages, respectively. All units of the proposed TDC are designed on-chip and implemented using built-in components of the FPGA board. Experimental results demonstrate superiority of the proposed $\Delta\sum$ TDC in terms of dynamic range, time-resolution and FoM over previous works which make it possible to employ this work in applications such as ADPLLs, range finders and ToF systems. It is worth noting that higher-order TDCs can be attained utilizing the proposed structure by increasing the number of cascaded stages, which results in higher SNR.

**Author Contributions:** All authors contributed to the present paper as follows. Conceptualization, A.M.Z.K. and E.F.; methodology, A.M.Z.K.; software, A.M.Z.K.; validation, E.F., S.H.M.A. and M.O.; formal analysis, A.M.Z.K.; investigation, A.M.Z.K.; resources, M.O.; data curation, A.M.Z.K.; writing—original draft preparation, A.M.Z.K.; writing—review and editing, S.H.M.A.; visualization, A.M.Z.K.; supervision, E.F.; project administration, E.F.; funding acquisition, S.H.M.A. and M.O.

**Funding:** This research was funded by Universiti Kebangsaan Malaysia, grant number DCP-2017-006/2.

**Acknowledgments:** The authors would like to thank Institute of Microengineering and Nanoelectronics (IMEN), University Kebangsaan Malaysia (UKM) for FPGA board and measuring devices. This research was partially supported by Universiti Kebangsaan Malaysia, Grant Code DCP-2017-006/2.

**Conflicts of Interest:** The authors declare no conflict of interest.

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
