# Peer review of "An FPGA-Based 16-Bit Continuous-Time 1-1 MASH ΔΣ TDC Employing Multirating Technique"

_electronics, doi:10.3390/electronics8111285_

Round 1
Reviewer 1 Report
The paper descripes the FPGA-based 1-1MASH delta sigma TDC employing multirating technique. The purpose of the design is to achieve high accuracy TDC for ToF-applications as you stated at the beginning and the end of the paper. You are also presenting some other applications areas at the beginning of the paper. The paper present the used technology quite clearly and also the idea of multirating in this specific technology, but there are some issues that has to be explained more carefully. Introduction should clearly explain the aimed application and what are the main performance parameters related to that application that should be improved. If you are not aiming any specific application, then you should explain that we are improving some parameter with some novel idea in the field of delta-sigma TDCs. Now the introduction is little bit confusing.
You are presenting time-to-digital converter and I want to see some results related to time-to-digital conversion. You are presenting just the measured output spectrums which are showing the noise shaping performance and also the effect of a multirating technique on the signal-to-noise ratio. Normally TDC papers present at least linearity curves (integral nonlinearity INL) and single shot precision as a function of an input time interval. If the aim is to design a TDC for ToF-laser range finder the single-shot precision is the one of the key performance parameter. This means that we should get very accurate result by using a single laser pulse with the wide input time range. Normally, basic TDCs based on Nutt-method having different interpolating structures can have nowadays the single shot precision of approximately 1 ps within the dynamic range several hundreds of nanoseconds. The precision of these TDCs can improved by averaging measurement results and with 10 000 measurements the precision is already 1ps/SQRT(10000) 0.01ps. In your case the delta sigma TDC is also counting several input time intervals and thus you are getting the resolution of 27 ps (or is this single shot precision. In some publication they are giving also the measured single shot precision, for example your reference [17]) and your time range is only 4.5 ns corresponding to 0.675 m in distance. This is quite short dynamic range. This should be explained in more detail if the aim is design a TDC for ToF-application. At the end of the introduction you are saying that we present this FPGA-based TDC. The novelty of using FPGA circuit compared to a fully integrated TDC circuit should be explained in more detail. In many application the size of the whole device is the key issue but FPGA board used here is quite large device. Could you comment on this issue too. Section 2.2. At the beginning of this section it could be good idea to explain clearly what are the improvement of the multirating technique before explaining it in more detail Section 2.1. GSRO-TDC formula (1). Is that correct. I am thinking that you should have the second-order noise-shaped quantization noise of the second stage fQ2, but you have fQ1 in your formula. Section 2.2. Multirating…. You have fs2 = m.fs1. You should use the multiplication symbol there. Section 4.1.GSRO.. You are explaining the operation of Fig. 7 (b) and you are saying that While either QEN or Qin are high output is fmax. I am understanding this so that you will get the fmax at the output when QEN = 1 and Qin can be 0 or 1 or Qin = 1 and QEN can be 0 or 1, but the circuit will pass through the fmax when both QEN = 1 and Qin=1. Section 5. Figure 10. What is the input signal in these measurements and the time difference between start and stop. These should be given. I would like to see some explanation related to averaging in presented work. Do you get the high precision result with a single input pulse or not. If you need several pulses, how many pulses should be used to get proper single-shot precision.Author Response
Point 1: Introduction should clearly explain the aimed application and what are the main performance parameters related to that application that should be improved? If you are not aiming any specific application, then you should explain that we are improving some parameter with some novel idea in the field of delta-sigma TDCs. Now the introduction is little bit confusing.
Response 1: Introduction of the paper has been modified.
Point 2: You are presenting time-to-digital converter and I want to see some results related to time-to-digital conversion. You are presenting just the measured output spectrums which are showing the noise shaping performance and also the effect of a multirating technique on the signal-to-noise ratio. Normally TDC papers present at least linearity curves (integral nonlinearity INL) and single shot precision as a function of an input time interval. If the aim is to design a TDC for ToF-laser range finder the single-shot precision is the one of the key performance parameter. This means that we should get very accurate result by using a single laser pulse with the wide input time range.
Response 2: The main aim of this work is introducing the concept of employing multirating technique in a ∆∑ GSRO-TDC for the first time. The proposed design takes advantage of employing GSRO guantizer to suppress noise leakage and at the same time benefits from multirating technique to enhance SNR. In fact, multirating technique does not affect linearity and just enhances the SNR. Thus, output spectrum (dynamic test) has been performed to show SNR and order of noise shaping of the proposed TDC to demonstrate that the main target has been met. Of course employing the proposed design in ToF and range finder systems needs linearity test, but we only suggest (not claim) that this work can be employed in applications such as ToF systems according to experimental results (SNR, dynamic range, FoM and time-resolution). Again, I emphasize that this paper tries to show this concept that employing multirating technique in a ∆∑ GSRO-TDC results in improvement of SNR, time-resolution and FoM. This point has been described explicitly in modified introduction.
Point 3: Normally, basic TDCs based on Nutt-method having different interpolating structures can have nowadays the single shot precision of approximately 1 ps within the dynamic range several hundreds of nanoseconds. The precision of these TDCs can improved by averaging measurement results and with 10 000 measurements the precision is already 1ps/SQRT(10000) 0.01ps. In your case the delta sigma TDC is also counting several input time intervals and thus you are getting the resolution of 0.27 ps (or is this single shot precision. In some publication they are giving also the measured single shot precision, for example your reference [17]) and your time range is only 4.5 ns corresponding to 0.675 m in distance. This is quite short dynamic range. This should be explained in more detail if the aim is design a TDC for ToF-application.
Response 3: Actually, 0.27 ps is time-resolution (i.e. the shortest time interval that TDC can realize and convert) of the proposed TDC which is the superior result rather previous works as can be seen in Table 1.
About Trange of our work and other references for example [17], I should say that according to time-resolution below 1 ps these TDCs are very accurate and they are used when accuracy is very important. They should be tailored according to application in which they are incorporated.
Also, dynamic range in these TDCs is defined as I noted in footnote of the Table 1 (DR = 20 log10 (Trange,rms / Tint,rms)) which relates dynamic range to the measured integrated noise to show immunity of TDC against noise to realize and convert even short time intervals. As can be seen this work shows superior DR rather previous works.
Point 4: At the end of the introduction you are saying that we present this FPGA-based TDC. The novelty of using FPGA circuit compared to a fully integrated TDC circuit should be explained in more detail. In many application the size of the whole device is the key issue but FPGA board used here is quite large device. Could you comment on this issue too.
Response 4: An explanation about using FPGA has been added to introduction.
Point 5: Section 2.2. At the beginning of this section it could be good idea to explain clearly what the improvement of the multirating technique are before explaining it in more detail.
Response 5: An explanation about advantages of Multirating technique has been added to section 2.2.
Point 6: Section 2.1. GSRO-TDC formula (1). Is that correct. I am thinking that you should have the second-order noise-shaped quantization noise of the second stage fQ2, but you have fQ1 in your formula.
Response 6: The Equation (1) has been corrected.
Point 7: Section 2.2. Multirating…. You have fs2 = m.fs1. You should use the multiplication symbol there.
Response 7: Multiplication symbol has been used.
Point 8: Section 4.1.GSRO.. You are explaining the operation of Fig. 7 (b) and you are saying that while either QEN or Qin are high output is fmax. I am understanding this so that you will get the fmax at the output when QEN = 1 and Qin can be 0 or 1 or Qin = 1 and QEN can be 0 or 1, but the circuit will pass through the fmax when both QEN = 1 and Qin=1.
Response 8: The description has been modified.
Point 9: Section 5. Figure 10. What is the input signal in these measurements and the time difference between start and stop. These should be given. I would like to see some explanation related to averaging in presented work. Do you get the high precision result with a single input pulse or not. If you need several pulses, how many pulses should be used to get proper single-shot precision.
Response 9: A description of the measurement setup has been presented in section 5 and figure 10 has been added to illustrate it.

Reviewer 2 Report
This paper presents a new TDC. The proposed TDC was implemented on an Altera Stratix IV FPGA board. I noticed that in Table.1 the comparison not fair because the other references (12, 13, 14, 15, and 17) are all implemented in different nm technology. The proposed work was performed on 40 nm, which is more modern than the rest. You have to run all the references using the same technique for better comparisons.
Author Response
Point 1: I noticed that in Table.1 the comparison not fair because the other references (12, 13, 14, 15, and 17) are all implemented in different nm technology. The proposed work was performed on 40 nm, which is more modern than the rest. You have to run all the references using the same technique for better comparisons.
Response 1: A comparison of this work and state-of-the-art TDCs using 65nm and 40nm technology in the recent 3 years has been added.

Reviewer 3 Report
This paper presents an FPGA-based 16-bit continuous-time 1-1 MASH ΔΣ TDC employing multirating technique. Some concerns are listed.
The authors should give a comparison of this work and the star-of-the-art, especially in the recent 3-years. How to measure the results? The authors should address the measurement setup and the used instruments. Figures are unclear, please redraw them.
Author Response
Point 1: The authors should give a comparison of this work and the star-of-the-art, especially in the recent 3-years. How to measure the results?
Response 1: A comparison of this work and state-of-the-art TDCs in the recent 3 years has been added.
Point 2: The authors should address the measurement setup and the used instruments.
Response 2: A description of the measurement setup has been presented in section 5 and figure 10 has been added to illustrate it.
Point 3: Figures are unclear, please redraw them.
Response 3: Figures were redrawn.

Round 2
Reviewer 1 Report
Corrections have been made and I think that the paper can be accepted for publication.
Reviewer 3 Report
The authors addressed all of my concerns.